# Carbon Nanotube-Based Printed All-Organic Microelectrode Arrays for Neural Stimulation and Recording

**DOI:** 10.3390/mi15050650

**Published:** 2024-05-14

**Authors:** Tatsuya Murakami, Naoki Yada, Shotaro Yoshida

**Affiliations:** Department of Electrical, Electronic, and Communication Engineering, Graduate School of Science and Engineering, Chuo University, Tokyo 112-8551, Japan; a19.7tfr@g.chuo-u.ac.jp (T.M.);

**Keywords:** microelectrode array, carbon nanotube, printing, neural stimulation, neural recording

## Abstract

In this paper, we report a low-cost printing process of carbon nanotube (CNT)-based, all-organic microelectrode arrays (MEAs) suitable for in vitro neural stimulation and recording. Conventional MEAs have been mainly composed of expensive metals and manufactured through high-cost and complex lithographic processes, which have limited their accessibility for neuroscience experiments and their application in various studies. Here, we demonstrate a printing-based fabrication method for microelectrodes using organic CNT/paraffin ink, coupled with the deposition of an insulating layer featuring single-cell-sized sensing apertures. The simple microfabrication processes utilizing the economic and readily available ink offer potential for cost reduction and improved accessibility of MEAs. Biocompatibility of the fabricated microelectrode was suggested through a live/dead assay of cultured neural cells, and its large electric double layer capacitance was revealed by cyclic voltammetry that was crucial for preventing cytotoxic electrolysis during electric neural stimulation. Furthermore, the electrode exhibited sufficiently low electric impedance of 2.49 Ω·cm^2^ for high signal-to-noise ratio neural recording, and successfully captured model electric waves in physiological saline solution. These results suggest the easily producible and low-cost printed all-organic microelectrodes are available for neural stimulation and recording, and we believe that they can expand the application of MEA in various neuroscience research.

## 1. Introduction

Microelectrode array (MEA) is one of the important technologies for measuring neural electrical signals, such as electroencephalography (EEG) [1] and electrocorticography (ECoG) [2], as well as for measuring in vitro neural signal for drug screening assays and neuroscientific investigations [3]. Conventionally, expensive precious metals such as gold (Au) and platinum (Pt) have been utilized as electrode materials; however, their utilization is accompanied by cost-intensive manufacturing processes and environmental concerns. Recently, organic nanomaterials such as carbon nanotubes (CNTs) and other carbon allotropes have been attracting attention as electrode materials due to their environmental friendliness, superior electrical conductivity, stability, and large specific surface area [4,5,6]. CNTs establish an enhanced electrode-cell interface for signal transmission compared to conventional metal electrodes, thereby augmenting the critical parameters of neural recording and stimulation such as signal-to-noise ratio (SNR) and charge injection density [7,8,9]. Furthermore, CNTs can be readily dissolved into ink and processed through simple and economic methods such as inkjet printing, microcontact printing and intaglio contact printing [9,10,11]. To date, numerous studies on carbon-based electrodes have emerged, including wearable electrodes for skin surface application [12], EEG recording within the ear [1], implantable electrodes for neural recording [13,14] and stimulation [15,16], facilitation of nerve regeneration [17], ECoG recording on brain surfaces [2], and electrophysiological investigation involving cultured neural cells [18,19]. Specifically, MEAs offer a high spatial resolution as well as remarkable sensitivity during neural recording and stimulation in both in vivo and in vitro environments [3], including on the brain surfaces [20,21,22,23,24,25,26] and in the brain tissue [27,28,29,30].

Various materials have been investigated for MEAs placed on the brain surface, including metal electrodes modified with organic materials for insulation [20], those utilizing organic materials to enhance electrical properties [21,22], as well as electrodes composed entirely of organic materials such as polypyrrole [23], poly(3,4-ethylenedioxythiophene):poly(styrene sulfonate) (PEDOT:PSS) [24,31,32], and graphene [25,26]. Similarly, metal MEAs intended for insertion into brain tissue have been modified with organic materials to improve insulation [27] or electrical properties [28,29], and some MEAs possess components for both surface placement and tissue insertion [30]. For in vitro electrophysiology, MEAs commonly have planar metal microelectrodes modified with organic materials including CNTs [33,34], composites of CNTs and conductive polymers [35], PEDOT:PSS [36], PEDOT:PSS and carboxylated graphene (cGO/PEDOT:PSS) [37], composites of PEDOT with carbon nanofibers (PEDOT:CNF) [38], and three-dimensional CNT microstructures [39]. Notably, there have been few reports of MEAs for in vitro experiments entirely composed of organic materials [40]. These metal-based MEAs partially modified with organic materials have demonstrated outstanding characteristics for neural stimulation and recording; however, challenges persist due to the high costs associated with precious metal materials and their fabrication processes, along with environmental concerns. Commercially available, metal-based MEAs have been prohibitively expensive, making them difficult to obtain and utilize casually in neuroscience studies. While MEAs composed entirely of organic materials offer the advantage of cost-effective production [23,24,25,26,40], none have yet fulfilled the requirement for sensor area dimensions of 10–20 µm essential for high-resolution electrophysiological experiments at the single-cell-level.

Here, we propose a simple and cost-effective method for fabricating microelectrodes using only organic materials: composite ink of CNT and paraffin wax (Figure 1). The CNT/paraffin composite ink has several remarkable characteristics as an electrode material: (i) ease of preparation by simply mixing the two materials [11], (ii) paraffin transitioning into a low-viscosity liquid above its melting point (i.e., approximately 60 °C) that facilitates easy pouring into a template mold and removal of excess ink by rubbing, (iii) paraffin solidifying at room temperature and maintaining a stable solid form during cell culture at 37 °C, (iv) the ability to undergo multiple cycles of solidification and liquefaction simply by changing the temperature that allows reshaping for precise patterning, and (v) high conductivity due to the presence of CNT. As depicted in Figure 1a, the device comprises a sensor area based on CNT/paraffin (locally exposed to approximately 20 × 20 µm), a wiring (250 or 500 µm × 12 mm), and a contact pad (for connection to electrical stimulation/recording equipment). The entire CNT/paraffin electrode is electrically insulated by a SU-8 layer, except for the locally exposed sensor area, which is of a similar size to a single neuron: 10–20 µm. The local exposure of the sensor area within the insulating layer reduces crosstalk between electrodes when forming the electrode array, facilitating high spatial resolution neural stimulation and recording [3].

To fabricate the microelectrode, SU-8 micropatterns were initially fabricated on a glass via photolithography, and subsequently, PDMS molds were formed utilizing these patterns. Finally, template molds for the electrode were fabricated by intaglio contact printing of SU-8 using poly(dimethylpolysiloxane) (PDMS) (Figure 1b). This technique allows for the rapid and cost-effective mass production of the template molds for electrodes. The CNT/paraffin composite ink was introduced into the mold and excess ink was removed by rubbing. The CNT/paraffin ink can be precisely micropatterned, and ink usage can be minimized compared to micro-contact printing [10]. A SU-8 insulation layer was then formed over the electrode and photolithographically patterned using a photomask to create the single-cell-sized, locally exposed aperture. The overall printing-based fabrication process with minimized photolithography is cost-effective since the electrodes can be manufactured using readily available organic materials without the need for precious metals, and the mold for intaglio contact printing can be reused multiple times after cleaning.

In this study, we focused on the development of the fabrication process and therefore fabricated a single electrode as a proof-of-concept rather than electrode arrays; however, in principle, it is feasible to array multiple electrodes using the same techniques. Thus, we believe that the proposed method is valuable for manufacturing low-cost MEAs that can be easily obtained and utilized in neuroscience studies. To demonstrate the utility of the fabricated device for neural stimulation and recording, we conducted microscopic observations of the electrode’s surface microstructures, biocompatibility tests by culturing neural cells, the evaluation of electrical stimulation performance through the measurement of electric double layer capacitance, and the assessment of electrical recording performance by measuring the electric impedance of the electrode and capturing the model electric signals using the electrode in a physiological saline solution.

## 2. Materials and Methods

### 2.1. Fabrication of the Microelectrodes

#### 2.1.1. Preparation of Photomasks

Electrodes were designed using computer-aided design software (Rhinoceros 7, Robert McNeel & Associates, Seattle, WA, USA), and two types of photomasks were fabricated using a maskless exposure system (Nano System Solutions, Okinawa, Japan). One photomask was designed for creating a template mold of the CNT/paraffin electrode, while the other was intended for patterning local exposure on the SU-8 insulation layer. Alignment marks were designed on the both photomasks to ensure precise alignment between the electrode and the locally exposed aperture of the SU-8 layer. Subsequently, the masks were developed using NMD-3 2.38% (Tokyo Ohka Kogyo, Kanawaga, Japan) and chrome etchant (Japan Chemical Industry, Shizuoka, Japan), followed by ultrasonic cleaning with acetone (FUJIFILM Wako Pure Chemical Corporation, Osaka, Japan).

#### 2.1.2. Preparation of Template Molds for the Electrodes

A 5 × 5 cm piece of SU-8 3045CF sheet (45 µm in thickness, Nippon Kayaku, Tokyo, Japan) was transferred onto a Si wafer (3 inch, 380 µm in thickness, Global Top Chemical, Tokyo, Japan) and heated to 40 °C on a hot plate (HP-2SA, AS ONE, Osaka, Japan). Two or three SU-8 sheets were stacked on the Si wafer to achieve thicknesses of 90 µm or 135 µm, respectively. Ultraviolet (UV) light was then exposed to the SU-8 using a mask aligner (PEM-800, Union, Tokyo, Japan) without removing the top protective film applied by the manufacturer. After exposure and post-baking on a hot plate at 55 °C for 5 min and 95 °C for 10 min, the protective film was peeled off, and the SU-8 was developed in propylene glycol monomethyl ether acetate (PGMEA, Tokyo Chemical Industry, Tokyo, Japan) and hard-baked in a vacuum dryer (ETTAS, AS ONE) at 180 °C for 60 min.

The Si wafer with SU-8 micropatterns were placed in a petri dish and molded with PDMS prepolymer (a mixture of monomer and curing agent at a mass ratio of 10:1, Silicone Elastomer Kit SYLGARD184, Dow, Midland, MI, USA). It was then degassed in a vacuum desiccator (VL, AS ONE), and heated on a hot plate at 75 °C for 1.5 h to cure the PDMS. The template mold was peeled off from the cured PDMS, and the PDMS mold was trimmed to leave 3 mm around the electrode patterns. The PDMS mold and a glass slide were then ultrasonically cleaned with 2-propanol (IPA) followed by pure water, and with acetone, IPA followed by pure water, respectively, and completely dried.

For intaglio contact printing of SU-8 onto a glass, a drop of SU-8 3010 (Nippon Kayaku) was casted in the PDMS mold and spread evenly across its surface. The glass slide was brought into contact with one side of the mold, gently pressed to adhere closely to the entire mold, and excess SU-8 was squeezed out. The SU-8 was cured by UV exposure followed by heating at 100 °C for 30 min on a hot plate. Once the PDMS mold was removed, the device was washed with ethanol (FUJIFILM Wako Pure Chemical Corporation).

#### 2.1.3. Fabrication of the Organic Microelectrode Utilizing the Template Molds

CNT/paraffin composite ink was prepared by mixing multi-walled carbon nanotubes (MWCNTs, Shenzhen FAYMO Technology, Shenzhen, China, MCN4301: 6–10 nm in diameter, 30–60 µm in length, MCN2101: 10–20 nm in diameter, 5–10 µm in length) and heated paraffin wax (ASTM D 87, melting point: 53–58 °C, Sigma-Aldrich, St. Louis, MO, USA) on a hot plate at 100 °C. The mixture was stirred thoroughly with a metal spatula for at least 1 min. Ink with 5 wt% MCN4301 was used for the electrode body due to its high conductivity, while ink with 7 wt% MCN2101 was used for the alignment marks owing to its relatively high viscosity and processability. The ink was stored at room temperature and reheated on a hot plate at 100 °C, followed by thorough stirring before use.

To fabricate the CNT/paraffin electrodes, the template molds and CNT/paraffin composite ink were heated on a hot plate at 100 °C, and the ink was poured into the template molds. Subsequently, the ink was allowed to cure at room temperature, and excess ink that protruded from the mold was removed by rubbing with a wooden stick or metal spatula. Conductivity of the electrode was checked by a digital multimeter (DT4282, HIOKI E. E. Corporation, Nagano, Japan), and if no conductivity was detected, the ink was reheated, additional ink was applied, and the curing process was repeated. Once conductivity was confirmed, the electrodes were washed with ethanol.

#### 2.1.4. Fabrication of the Insulation Layer with Locally Exposed Apertures

A 2 × 2.5 cm piece of SU-8 3045CF was adhered to the top of the CNT/paraffin electrode on a hot plate heated to 40 °C. The alignment marks of the photomask were aligned with the alignment marks created by CNT/paraffin using a mask aligner. After UV exposure, the whole device was post-baked at 55 °C for 3 min and 95 °C for 5 min on the hot plate. Subsequently, the device was allowed to cool to room temperature, and then the top protective film of SU-8 3045CF was peeled off before development in PGMEA.

### 2.2. Microscopic Evaluation of the Microelectrodes

Three-dimensional (3D) shapes and microscopic surface of the fabricated electrodes were observed using a laser microscope (VK-9710, KEYENCE, Osaka, Japan) and a scanning electron microscope (SEM, VE-8800, KEYENCE). Laser microscopy was employed to conduct 3D cross-sectional measurements from the bottom to the top of the fabricated electrodes with a measurement pitch of 0.5 µm and a 20× objective lens, following the manufacturer’s setting of “transparent body mode: top surface” that allowed for precise scanning of the electrode materials. SEM observation of the electrode was performed without any surface treatment, at an acceleration voltage of 1 kV.

### 2.3. Cell Culture and Biocompatibility Test

#### 2.3.1. Cell Culture

PC12 neuron-like cells (91A001, CH3 Bio Systems, New York, NY, USA) were cultured according to standard protocols as a model of neural cells. The cells were previously maintained in Dulbecco’s Modified Eagle Medium (Advanced DMEM, 12491015, Thermo Fisher Scientific, Waltham, MA, USA) containing 10% (*v*/*v*) fetal bovine serum (FBS, S-FBS-NL-015, Serana Europe, Brandenburg, Germany), 2% (*v*/*v*) L-glutamine (Sigma-Aldrich), and 1% (*v*/*v*) penicillin-streptomycin (Sigma-Aldrich) in a humidified 37 °C incubator with 5% CO_2_ (MCO-5ACUV-PJ, PHC, Tokyo, Japan), and passaged at a 1:3 ratio using trypsin-EDTA (Sigma-Aldrich) every 3 d.

When culturing PC12 cells on the fabricated electrodes, the electrodes were treated with O_2_ plasma using a plasma processing device (PC-400, STREX, Osaka, Japan) for 20 s and then washed using serum-free Advanced DMEM containing 100 ng mL^−1^ of nerve growth factor (NGF, 2.5S from murine submaxillary gland, Sigma-Aldrich), 2% (*v*/*v*) L-glutamine (Sigma-Aldrich) and 1% (*v*/*v*) penicillin-streptomycin for nerve differentiation. The device was placed in a 60 mm diameter culture dish filled with the serum-free culture medium and sterilized with a UV lamp (LUV-16, AS ONE). PC12 cells were collected using trypsin-EDTA (Sigma-Aldrich) and seeded on the electrodes, then incubated for 2 d.

#### 2.3.2. Biocompatibility Test

After incubation for 2 d, the cells in a culture dish containing the fabricated electrodes or those in a culture dish without electrodes were stained using a live or dead cell viability assay kit (22789, AAT Bioquest, Pleasanton, CA, USA) that contained calcein-AM for staining live cells as green, and propidium iodide (PI) for staining the nuclei of dead cells as red. The nuclei of cells were co-stained with Hoechst 33342 (H342, Dojindo, Kumamoto, Japan) as blue for counting the number of cells. The dyes were supplemented in the culture medium following the manufacture’s protocol, and the cells were incubated for 1 h. Without replacing the culture medium with dye to prevent washing out detached dead cells, the fluorescence from the stained cells were observed using a phase contrast/fluorescent microscope (CKX53, EVIDENT, Tokyo, Japan). While observing under the phase contrast microscope, randomly selected cells in the culture dish containing the fabricated electrodes or in the culture dish without electrodes (control) were photographed, and their fluorescence images were captured by a camera. To quantify the cell viability, the number of cells in the images were counted manually from the blue fluorescence, while simultaneously counting whether the cells were emitting green or red fluorescence. Cell viability was determined by dividing the number of cells emitting green fluorescence by the total number of cells, with a double check to ensure absence of red fluorescence. Note that the red fluorescence was pseudo-colored as magenta using software (ImageJ 1.53e) for visibility.

### 2.4. Characterization for Neural Stimulation

To evaluate the performance of the fabricated electrode in neural electrical stimulation compared to conventional metal electrode, electric double layer capacitance and occurrence of bubbles due to water electrolysis during electrical stimulation was investigated, ensuring no cytotoxic Faradaic currents were generated with the fabricated electrode.

To fabricate electrodes for electrical double-layer capacitance measurement, a frame of approximately 2 cm × 2 cm was formed on glass using SU-8 3010. Following the same procedure used for electrode fabrication, CNT/paraffin composite ink was poured into the frame, cured, and then immersed in PGMEA for 5 min and subsequently washed with ethanol. The fabricated electrode, along with a conventional Au electrode, was immersed in physiological saline solution (pure water containing 0.9 wt% NaCl) with a Pt counter electrode (CE-8, EC FRONTIER, Kyoto, Japan) and a silver/silver chloride (Ag/AgCl) reference electrode (RE-T1A, EC FRONTIER) and connected to an electrochemical analyzer (ECstat-301, EC FRONTIER). Cyclic voltammetry was performed at a scanning speed of 20 mV/s from 0 V to 0.8 V (vs Ag/AgCl). For the Au electrode, cyclic voltammetry was performed with a 1 MΩ resistor connected between the Au electrode and the electrochemical analyzer to prevent measurement error caused by excessive current flowing into the electrochemical analyzer. The electric double layer capacitance of the fabricated electrode was calculated as the current value at 0.4 V divided by the scanning speed.

To evaluate the occurrence of bubbles during electrical stimulation, the electrode with a locally exposed insulation layer as described in the Section 2.1 as well as a conventional Pt electrode were employed. The fabricated electrode or Pt electrode were connected to the positive terminal of a function generator (AFG-2005, GWINSTEK, Taipei, Taiwan), while a Pt counter electrode was connected to the negative terminal. The electrodes were immersed in a physiological saline solution, and a 1 Hz, 5 Vpp (peak-to-peak voltage) sinusoidal electrical signal was applied from the function generator. The occurrence of bubbles was observed as the voltage was increased to 10 Vpp.

### 2.5. Characterization for Neural Recording

For electrical impedance measurement, the fabricated electrode as described in Section 2.1 was connected to one probe of the LCR meter (IM3536, HIOKI) and immersed in physiological saline along with the other probe. The absolute value of the electrical impedance of the electrode at 1 kHz and 0.5 Vpp was repeatedly measured for 10 times, and the average value was calculated. Similarly, the impedance measurements were conducted for the electrode subjected to 20 s of O_2_ plasma treatment and for the electrode completely insulated with no apertures. The impedance value was normalized by multiplying the impedance value by the apparent exposed area of 15.3 × 22.2 = 340 µm^2^ that was measured as Section 2.2 and in Figure 2e.

For electrical signal recording, a sinusoidal electrical signal of 1 kHz and 1 Vpp was applied in a physiological saline solution through a Pt electrode connected to a function generator. The fabricated electrode was connected to an oscilloscope (RTA4004, ROHDE&SCHWARZ) and immersed in the saline to capture the flowing electrical signal. The same electrical signals were also acquired using the bare CNT/paraffin electrode without an insulating layer and conventional Ag/AgCl metal electrodes.

## 3. Results and Discussion

### 3.1. Fabrication and Microscopic Observation of the All-Organic Microelectrodes

Figure 2a shows the SU-8 template mold for the CNT/paraffin electrode fabricated by the intaglio contact printing method using a PDMS mold. The printing method, requiring only a reusable PDMS mold and a UV lamp, enabled rapid and low-cost fabrication within minutes, contrasting with the conventional photolithography technique. During SU-8 printing, tiny air bubbles that tended to be introduced were removed when pressing the glass against the PDMS mold, ensuring a bubble-free sensor area as depicted in Figure 2a.

Figure 2b shows the fabricated CNT/paraffin electrode with the SU-8 insulation layer printed using the template mold. To establish the sensor area on the CNT/paraffin electrode, a locally exposed aperture within the SU-8 insulation layer was formed by an alignment technique detailed in Section 2.1. As previously demonstrated [11], it was possible to form CNT/paraffin conductive micropatterns directly by intaglio contact printing. However, the relatively thick CNT/paraffin micropatterns (10 µm or more) and their large surface roughness frequently led to partial breaking of the insulation layer at the edge of the electrodes. Therefore, we developed a method in which a template mold was made by intaglio contact printing first, and then CNT/paraffin ink was poured into it. This method allowed for the elimination of the gaps at the edges of the CNT/paraffin, enabling the uniform formation of the insulation layer without partial breakage.

The rapidly phase-changeable, thermoplastic nature of the CNT/paraffin composite allowed for the rapid fabrication of electrodes within minutes by simply pouring heated ink into the template mold and subsequently stopping the heating. Moreover, when the CNT/paraffin overflowed from the mold or if the ink quantity was insufficient, reheating enabled the CNT/paraffin to quickly revert to a liquid state and allowed for easy removal of excess material or addition of more ink. Furthermore, hardened CNT/paraffin could be easily shaved with metal spatulas, facilitating precise aligning of the height of the contact pads to match the mold and ensuring ease of attachment of connecting wires for electrical stimulation and measurement devices.

These remarkable properties suggest the versatility of this method in creating electrodes of various designs, as demonstrated in this study where devices with electrode widths of 250 µm or 500 µm and electrode thicknesses (i.e., mold depth) of 90 µm or 135 µm were successfully fabricated. To achieve highly integrated electrode arrays, it will be necessary to reduce the width of the wiring, ideally to around single-cell-sized dimensions of 10 µm. Therefore, future efforts will focus on improving printing techniques including adjustment of ink viscosity to develop a fabrication process for smaller lines.

In order to confirm the presence of locally exposed single-cell-sized aperture in the insulating layer and to observe the surface nanostructures of the CNT/paraffin electrode, SEM observations were conducted. Figure 2c shows a SEM image of the locally exposed sensor area, revealing the presence of a single-cell-sized aperture with a diameter of 20 µm. Although the shape of the aperture exhibited slight distortion from the designed square, similar to the standard optimization process in conventional photolithography, it is anticipated that adjustments in light exposure levels and baking conditions of SU-8 could bring it closer to a square shape. Observation of the boundary between the SU-8 insulating layer and the contact pad (Figure 2d) revealed rough, porous surface nanostructures of the CNT/paraffin electrode. The significantly large specific surface area is presumed to primarily stem from the nanostructures of CNTs. As frequently observed in other MEA studies employing porous material modifications on metal electrodes [3], it is suggested that the porous surface of CNT/paraffin contributes to the reduction of electrode impedance, thereby implying its ability for high SNR neural recording. Detailed material characterizations regarding the surface structure of the CNT/paraffin composites have also been confirmed in a prior study [11].

To examine whether the locally exposed aperture opened on the insulating layer penetrated through to expose the underlying electrode, we observed the cross-sectional shapes of the sensor area and evaluated the size using laser microscopy (Figure 2e). It was confirmed that the surface of CNT/paraffin electrode was exposed for 15.3 µm × 22.2 µm, and the depth of the aperture was 36 µm, similar to the thickness of the SU-8 sheet (45 µm) used as the insulation layer. The measured depth was thinner than the nominal thickness of the insulating sheet (SU-8 3045CF) provided by its manufacturer. This discrepancy could be attributed to measurement errors in laser reflectance due to the transparency of SU-8, or it may have resulted from shrinkage of the SU-8 sheet when pressed onto the electrode during formation. On the other hand, the exposure of electrodes at the sensor area was consistently verified post-fabrication by measuring the electrical resistance between the exposed area treated with ethanol and the contact pad using a digital multimeter, ensuring a decrease in resistance. Additionally, as later discussed, confirmation of electrode exposure was achieved by comparing impedance measurements and electrical signal recordings in physiological saline to those obtained when electrodes were not exposed.

### 3.2. Cell Culture and Biocompatibility Test on the Microelectrodes

Since the objective of the electrode is stimulation and recording of cultured neurons, it is essential that the electrode exhibits no cytotoxicity and allows for culturing neural cells. To assess cytotoxicity, we cultured PC12 neuron-like cells as model neural cells in dishes containing the fabricated electrode and compared cell viability with those in control dishes. PC12 cells were cultured on the SU-8 insulating layer atop the CNT/paraffin microelectrode. Figure 3a shows a microscopic image of PC12 cells cultured on the device for 2 days. The cells adhered to the insulating layer on the electrodes and remained viable throughout the culture period. Importantly, no dissolution of paraffin was observed during the culture period, and the electrodes maintained stable shapes. This stability is attributed to the higher melting point of the paraffin used in this study (53–58 °C) compared to the commonly used neuronal culture temperature of 37 °C. As shown in Figure 3a, the transparent SU8 insulation layer is suitable for observation using phase-contrast microscopy and fluorescence microscopy commonly used in neuroscience. However, it was found that performing an observation of cells on CNT/paraffin electrodes, formed using molds with thicknesses of 90 µm or 135 µm, was not possible as they prevented light transmission. Future improvements will focus on reducing thickness, making the electrodes transparent by utilizing graphene or conductive polymers, and reducing the width of wiring as mentioned in Section 3.1 to facilitate easier cell observation.

In Figure 3b, the upper-left panel shows that PC12 cells cultured on dishes containing the device were healthy enough for extending neurites by neural differentiation induced by added NGF. Results of nuclear staining (blue) and live/dead assay staining (cytoplasm of live cells: green, nuclei of dead cells: red) conducted on these cells are also presented in Figure 3b. Note that red is pseudo-colored as magenta for visibility. The majority of cells were stained as green, suggesting the biocompatibility of the device. Cell viability was analyzed using the procedure outlined in Section 2.3.2, and the results were compared with those from control culture dishes without the device (Figure 3c). Cells cultured with electrodes exhibited a high cell viability of 95%, comparable to cells in control dishes, indicating the biocompatibility of the electrodes. Further robust statistical analysis on the biocompatibility will be conducted in future work.

### 3.3. Characterization for Neural Stimulation

To ensure minimal impact on the viability of neuronal cells during electrical stimulation using MEA, it is essential to avoid electrolysis of the culture medium, as well as generation of cytotoxic bubbles and pH fluctuations associated with the electrolysis. Organic porous electrodes with a large specific surface area and correspondingly significant electric double layer capacitance enable the safe stimulation of neuronal cells without electrolysis, bubble generation, or pH changes through non-Faradaic currents even when subjected to high voltages [24]. This capability arises from their capacity to store a considerable amount of electronic charge on the electrode surface. Therefore, we evaluated the electric double layer capacitance of the CNT/paraffin composites using cyclic voltammetry (CV). CV measurements were carried out on CNT/paraffin electrodes and Au electrodes in physiological saline solution, within the voltage range of 0 V to 0.8 V (vs Ag/AgCl) at a scan rate of 20 mV/s. As shown in Figure 4a, while the Au electrode exhibited linear behavior, the CNT/paraffin electrode showed a rectangular response indicating electrical charge accumulation, commonly observed in electrodes with large electric double layer capacitance [24]. Calculating the double layer capacitance at 0.4 V, within the region of constant current, yielded a value of 3.08 µF/cm^2^. This higher capacitance compared to conventional metal electrodes suggests reduced susceptibility to electrolysis during electrical stimulation. However, the capacitance of these electrodes was lower than those modified with conductive polymers on nanostructured carbon fibers (e.g., 74 mF/cm^2^ [24]). Therefore, future efforts will focus on increasing the specific surface area through further modifications with conductive polymers and nanostructures to develop safer electrodes.

The inhibitory effect on electrolysis and bubble generation due to the high electric double layer capacitance was confirmed by applying voltage to the electrodes in physiological saline solution (Figure 4b). A sinusoidal electrical wave of 1 Hz and 5 Vpp was generated by a function generator and applied to CNT/paraffin electrodes and conventional metal Pt electrodes immersed in physiological saline solution. As the voltage was gradually increased, bubbles were visibly observed at around 10 V on the Pt electrode, while no bubbles were generated under the same conditions on the CNT/paraffin electrode. When high voltage is applied to conventional metal microelectrodes during electrical stimulation of neuronal cells, the electrodes themselves may be ruptured by bubbles derived from electrolysis. These results suggest that the CNT/paraffin electrodes, with their high electric double layer capacitance, provide a safer and more stable method for electrical stimulation compared to conventional metal electrodes.

### 3.4. Characterization for Neural Recording

For neural recording, the reduction of electric impedance is necessary to increase the SNR of recorded electric signals [24]. To assess whether the fabricated electrodes had sufficiently low electric impedance for neural recording, we measured the absolute value of the impedance under 1 kHz and 0.5 Vpp voltage application using a LCR meter (Figure 5a). The impedance measurement at the frequency of 1 kHz is commonly utilized as a standard for evaluating the electrical characteristics of MEAs, since it represents a typical frequency in neural recording [33]. Similar to other studies [36], we focused on the impedance at 1 kHz rather than a wide range of frequency characteristics, making it easier to compare the impedance reduction of electrodes under different conditions. For comparison, impedance of a totally insulated electrode with no aperture in the insulation layer was measured as 53.5 MΩ, indicating the excellent insulation ability of the insulation layer. The impedance of the sensor area of the electrode with an insulation layer with locally exposed aperture was 17.4 MΩ (i.e., 59.1 Ω·cm^2^). The reduction in impedance resulted from the local exposure of the CNT/paraffin electrode through the aperture in the insulating layer. However, the impedance was insufficiently low for neural recording, which was attributed to the presence of the remaining bubbles inside the aperture upon introduction of the physiological saline solution that hindered ion movement. Therefore, measurements were also conducted under conditions where the entire electrode was made hydrophilic by O_2_ plasma treatment before introducing the physiological saline solution, making it less likely for bubbles to remain. As a result, the impedance was confirmed to be 734 kΩ (i.e., 2.49 Ω·cm^2^), which was comparable to impedance levels confirmed in previous studies to be sufficiently low for high-SNR neural recording [26]. These results suggest that the fabricated electrodes were useful for neural recording, and that hydrophilic treatment should be applied to the electrodes prior to the recording. As the plasma treatment is not commonly used in typical neuroscience experiments, in the future, we plan to attempt hydrophilization by molecular modification of the electrode surface.

Finally, to investigate the feasibility of neural recording using the fabricated electrodes, we applied model electrical signals mimicking neural activity at a frequency of 1 kHz into physiological saline solution and measured the signals using the electrodes (Figure 5b). For comparison, measurements were conducted using conventional Ag/AgCl metal electrodes, bare CNT/paraffin electrodes with no insulating layer, and CNT/paraffin electrodes with an insulating layer with a locally exposed aperture and subjected to hydrophilization treatment. Both the conventional Ag/AgCl metal electrodes and the bare CNT/paraffin electrodes demonstrated similarly high SNR during measurement. This result suggests that the CNT/paraffin electrodes themselves possess sufficiently low electrode impedance similar to conventional electrodes, implying the usefulness of these electrodes for neural recording. However, in the case of locally exposed CNT/paraffin electrodes with the insulating layer, although the waveform of the signal was measurable, it exhibited more noise compared to conventional electrodes or bare CNT/paraffin electrodes. The result indicates increased ion transfer resistance due to the deep depth of the locally exposed aperture (36 µm). Future improvements will focus on reducing the depth of the aperture and modifying the surface of the CNT/paraffin electrodes with organic polymers, allowing the electrode to be protruded to the surface of the insulating layer, thereby enabling neural recording with even higher SNR.

## 4. Conclusions

In this study, we reported a novel method for manufacturing cost-effective and simple MEAs for neural stimulation and recording using a CNT/paraffin composite material through a printing process. Compared to conventional MEAs based on precious metals, our approach lowers costs, thereby expanding the applicability of MEAs in drug screening and neuroscience experiments. Although we focused on assessing single electrode characteristics, it was suggested that an array of multiple electrodes could be fabricated without crosstalk since it was possible to form an insulating layer with single-cell-sized aperture atop the electrode.

The high biocompatibility of the electrodes was indicated by culturing neural cells, which showed comparable cell survival rates to conventional culture conditions. We found that CNT/paraffin electrodes reduced electrolysis and bubble generation during neural stimulation compared to conventional electrodes via measurements of electric double layer capacitance. Hydrophilization treatment of apertures improved electrode impedance, indicating potential for high-fidelity neural recording. Electrical recording experiments revealed that CNT/paraffin enabled recording with similarly high SNR as conventional metal electrodes.

Limitations of the current device include electrode width, opacity hindering cell observation, and aperture depth potentially leading to bubble entrapment. Future research will focus on optimizing printing conditions to reduce the electrode width, enhancing electrode transparency, and modifying surface properties to improve long-term stability.

## Figures and Tables

**Figure 1 micromachines-15-00650-f001:**
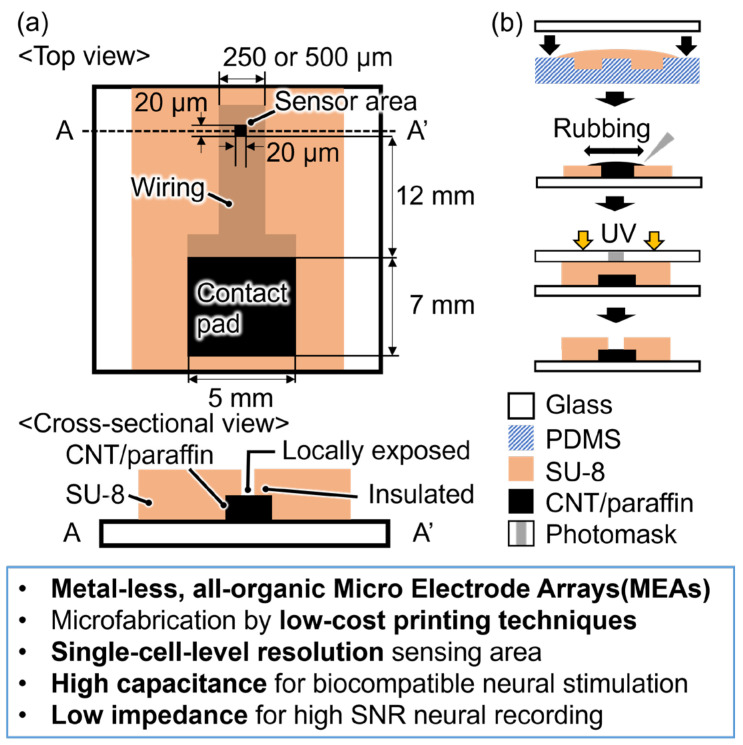
Conceptual illustration of the proposed all-organic microelectrode for in vitro neural stimulation and recording, and description of its advantages over conventional, commercially available metal-based microelectrodes. (**a**) Top view and cross-sectional view of the electrode showing its simple material composition (CNT/paraffin/SU-8), structure and dimensions of single-cell-sized high-resolution sensor area, wiring, and a contact pad for connection to neural stimulation/recording equipment. (**b**) Schematic of the simple and low-cost, printing-based fabrication process of the proposed microelectrode.

**Figure 2 micromachines-15-00650-f002:**
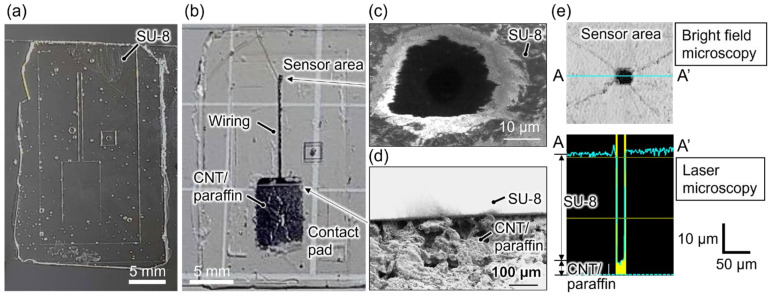
Photographs of (**a**) the SU-8 template mold and (**b**) printed CNT/paraffin electrode with SU-8 insulation layer on top with locally exposed sensor area. SEM images of (**c**) the sensor area and (**d**) the boundary between the SU-8 insulation layer and CNT/paraffin contact pad. (**e**) A bright field microscopic image (**top**) and laser microscopic cross-sectional view (**bottom**) of the sensor area.

**Figure 3 micromachines-15-00650-f003:**
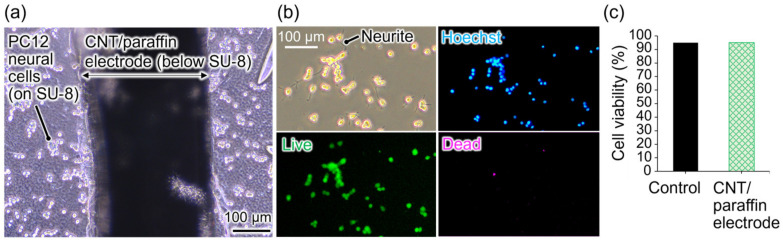
Biocompatibility test of the microelectrodes. (**a**) A phase contrast microscopic image of PC12 neural cells cultured on the fabricated electrodes showing high viability. Note that the cells are cultured on the transparent SU-8 insulation layer atop the CNT/paraffin electrode. (**b**) Phase contrast and fluorescent microscopic images of the PC12 cells stained by Hoechst 33342 (blue, nuclei), calcein-AM (green, live cells), and PI (magenta, dead cells, pseudo colored for visibility). (**c**) Cell viability of the PC12 cells cultured in a normal dish as a control (number of cells: 319) or in a dish with the CNT/paraffin electrode (number of cells: 502).

**Figure 4 micromachines-15-00650-f004:**
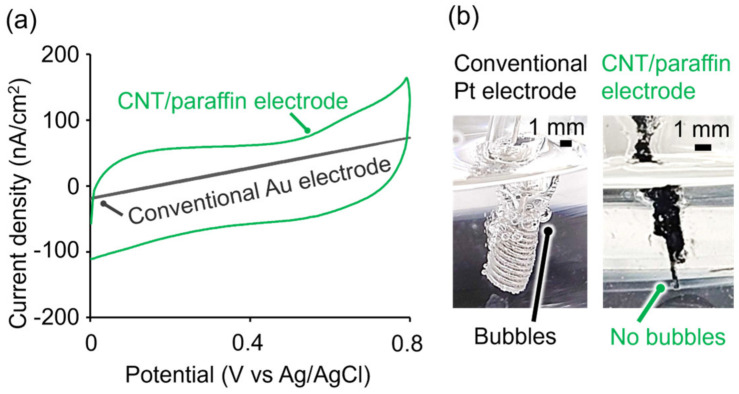
Characterization of the microelectrodes for neural stimulation. (**a**) Electric double layer capacitance measured by cyclic voltammetry in a physiological saline solution at scanning speed 20 mV/s from 0 V to 0.8 V vs. Ag/AgCl. Photographs of (**b**) conventional Pt electrode and CNT/paraffin electrode under electrical stimulation at 10 V in a physiological saline.

**Figure 5 micromachines-15-00650-f005:**
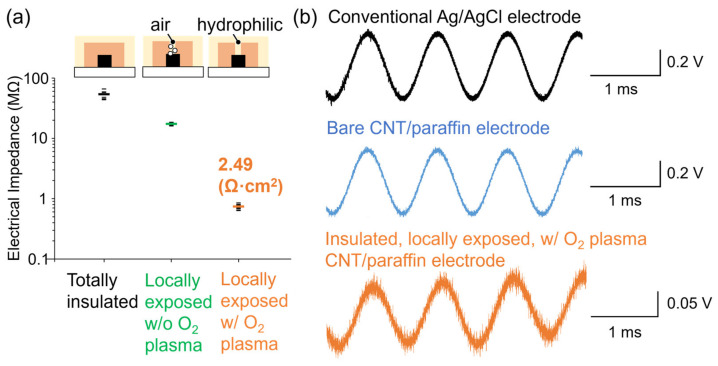
Characterization of the microelectrodes for neural recording. (**a**) Electrical impedance at the frequency of 1 kHz of three types of the microelectrode: totally insulated electrodes, electrodes with locally exposed insulation layer without O_2_ plasma treatment, and the electrodes with O_2_ plasma treatment. Dot plots show raw data (n = 10 for each type of electrodes) and bars shows their mean values. Width and depth of the electrode were 250 μm and 90 μm, respectively. (**b**) Recorded model sinusoidal electrical signals at the frequency of 1 kHz in physiological saline with conventional Ag/AgCl electrode (**top**), with CNT/paraffin electrode without insulation layer (**middle**), and with CNT/paraffin electrode with O_2_ plasma-treated locally exposed insulation layer (**bottom**).

## Data Availability

The original contributions presented in the study are included in the article, further inquiries can be directed to the corresponding author.

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
