# Peer review of "Carbon Nanotube-Based Printed All-Organic Microelectrode Arrays for Neural Stimulation and Recording"

_micromachines, 2024, doi:10.3390/mi15050650_

Round 1
Reviewer 1 Report
Comments and Suggestions for Authors
Yoshida and co-workers report a low-cost printing process of carbon nanotube (CNT)-based, all-organic microelectrode arrays (MEAs) suitable for in vitro neural stimulation and recording. This research is interesting and demonstrate a new method, however, introduction section, materials and methods and results and discussion sections should be revised according to the following comments. I invite the authors to carefully address these comments as the manuscript requires a major revision.
Specific comments:
1- Introduction page 1 “ Recently, organic nanomaterials such as carbon nanotubes (CNTs) and other car-33 bon allotropes have been attracting attention as electrode materials due to their environmental friendliness, superior electrical conductivity, stability, and large specific surface area “ I recommend that the authors add the following references:
a- M Eslamian, F Mirab, VK Raghunathan, S Majd, MR Abidian (2021) “Organic semiconductor nanotubes for electrochemical devices” Advanced functional materials 31 (49), 2105358
b- DR Kim, M Abidian, D Martin (2004) “Synthesis and characterization of conducting polymers grown in hydrogels for neural applications” Mat. Res. Soc. Symp. Proc. 1 F 5, 5.1-F5
2- Introduction page 2 “poly(3,4-ethylenedioxythiophene):poly(styrene sulfonate) (PEDOT:PSS) “ I recement that authors include the following papers by Abidian and co-workers and Majd and co-workrs on all organic electrodes made of PEDOT:PSS /PEGDA and PEDOT:PSS/PDMS respectively :
c- O Dadras‐Toussi, M Khorrami, ASC Louis Sam Titus, S Majd, C Mohan, and MR Abidian (2002) “Multiphoton lithography of organic semiconductor devices for 3D printing of flexible electronic circuits, biosensors, and bioelectronics” Advanced Materials 34 (30), 2200512
d- SH Park, G Yang, N Madduri, MR Abidian, S Majd (2014) “Hydrogel-mediated direct patterning of conducting polymer films with multiple surface chemistries” Advanced materials 26 (18), 2782
3- The authors only showed the impedance at 1kHz in figure 5, They should include impedance spectroscopy over the range of frequency for the data shown in figure 5a .
4- Data shown in figure 3c is only for on sample, Cell viability must be done with statistical analysis. The authors should include more samples for the biocompatibility evaluation
5- Scale bars should be added to Figure 4b
6- Data shown in Figure 5a with error bars must include the sample size (n) and standard deviation (SD) or standard error of mean (SEM) in the figure caption.
7- Statistical Analysis should be included in method section.
Comments on the Quality of English LanguageAuthor Response
Please see the attachment.

Reviewer 2 Report
Comments and Suggestions for Authors
In this paper, the authors report a low-cost printing process of carbon nanotube (CNT)-based, all-organic microelectrode arrays (MEAs) suitable for in vitro neural stimulation and recording
1- Figure 4b, please provide better-resolution images.
2- There is a fabrication and material. Before the biocompatibility test, I was expecting some material characterization like I-V curves, EDX, and FESEM of the electrode.
3- The conclusion is a bit long. Please reduce/improve it.
Comments on the Quality of English LanguageEnglish is ok
Round 2
Reviewer 1 Report
Comments and Suggestions for Authors
The authors addressed the reviewer's comments and I recommend this manuscript for publication after a minor revision for editing.
Comments on the Quality of English LanguageThere are few minor edits and corrections that need to be done. I recommend that the authors ask a native English speaker to proof-read the manuscript.